# Multiple origins of dorsal ecdysial sutures in trilobites and their relatives

Kun-sheng Du[1,2†], Jin Guo[2,3†], Sarah R Losso[4]*, Stephen Pates[5,6]*, Ming Li[7], Ai-lin Chen[1,8]*

[1]Research Center of Paleobiology, Yuxi Normal University, Yuxi, China; [2]Key Laboratory for Palaeobiology and MEC International Joint Laboratory for Palaeoenvironment, Institute of Palaeontology, Yunnan University, Kunming, China; [3]Management Committee of the Chengjiang Fossil Site World Heritage, Chengjiang, China; [4]Museum of Comparative Zoology and Department of Organismic and Evolutionary Biology, Harvard University, Cambridge, United States; [5]Centre for Ecology and Conservation, University of Exeter, Penryn, United Kingdom; [6]Department of Zoology, University of Cambridge, Downing Street, Cambridge, United Kingdom; [7]Institute of Geology, Chinese Academy of Geological Sciences, Beijing, China; [8]State Key Laboratory of Palaeobiology and Stratigraphy, Nanjing Institute of Geology and Paleontology, Nanjing, China

*For correspondence:
sarahlosso@g.harvard.edu (SRL);
s.pates@exeter.ac.uk (SP);
ailinchen@yxnu.edu.cn (AC)

†These authors contributed equally to this work

Competing interest: The authors declare that no competing interests exist.

**Abstract** Euarthropods are an extremely diverse phylum in the modern, and have been since their origination in the early Palaeozoic. They grow through moulting the exoskeleton (ecdysis) facilitated by breaking along lines of weakness (sutures). Artiopodans, a group that includes trilobites and their non-biomineralizing relatives, dominated arthropod diversity in benthic communities during the Palaeozoic. Most trilobites – a hyperdiverse group of tens of thousands of species - moult by breaking the exoskeleton along cephalic sutures, a strategy that has contributed to their high diversity during the Palaeozoic. However, the recent description of similar sutures in early diverging non-trilobite artiopodans means that it is unclear whether these sutures evolved deep within Artiopoda, or convergently appeared multiple times within the group. Here, we describe new well-preserved material of *Acanthomeridion*, a putative early diverging artiopodan, including hitherto unknown details of its ventral anatomy and appendages revealed through CT scanning, highlighting additional possible homologous features between the ventral plates of this taxon and trilobite free cheeks. We used three coding strategies treating ventral plates as homologous to trilobite-free cheeks, to trilobite cephalic doublure, or independently derived. If ventral plates are considered homologous to free cheeks, *Acanthomeridion* is recovered sister to trilobites, however, dorsal ecdysial sutures are still recovered at many places within Artiopoda. If ventral plates are considered homologous to doublure or non-homologous, then *Acanthomeridion* is not recovered as sister to trilobites, and thus the ventral plates represent a distinct feature to trilobite doublure/free cheeks.

## eLife assessment

The authors present 16 new well-preserved specimens from the early Cambrian Chengjiang biota. These specimens potentially represent a new taxon which could be **useful** in sorting out the problematic topology of artiopodan arthropods - a topic of interest to specialists in Cambrian arthropods. The authors provide **solid** anatomical and phylogenetic evidence in support of a new interpretation of the homology of dorsal sutures in trilobites and their relatives.

## Introduction

Euarthropods, more specifically members of the clade Artiopoda, dominated diversity in Palaeozoic animal communities (*Daley et al., 2018*). This group, which includes trilobites and their non-biomineralizing relatives, are united by a morphology consisting of a headshield covering a pair of uniramous antennae and at least three pairs of biramous limbs, followed by a dorsoventrally flattened body with a series of biramous appendages (*Stein and Selden, 2012*; *Giribet and Edgecombe, 2019*). The origins of key morphological features facilitating the diversity of trilobites can be found in their non-biomineralizing artiopodan relatives (e.g. *Du et al., 2019*; *Schmidt et al., 2022*). The cephalon, as an informative and functionally specialized body region, affords pivotal data for resolving the phylogenetic relationships and evolutionary history of arthropods, artiopodans, and trilobites in particular (e.g. *Stubblefield, 1936*; *Lieberman, 2002*; *Budd and Telford, 2009*; *Yang et al., 2013*; *Paterson et al., 2019*; *Strausfeld et al., 2022*).

The presence, shape, and character of sutures in the trilobite cephalon have been central to classification schemes of the group for nearly 200 years (e.g. *Emmrich, 1839*; *Salter, 1864*; *Beecher, 1897a*; *Beecher, 1897b*; *Stubblefield, 1936*; *Rasetti, 1945*; *Whittington et al., 1997*), while the similarity in the cranidial outlines of Cambrian suture-bearing groups supported a single origin or strong evolutionary constraint on its origins (*Foote, 1991*; *Hughes, 2007*). These sutures provided lines of weakness along which the cephalon split during moulting, creating an anterior ecdysial gape through which the animal exited the old exoskeleton (e.g. *Henningsmoen, 1975*; *Whittington, 1990*; *Daley and Drage, 2016*). Sutures may have served additional purposes (*Stubblefield, 1936*) and were under additional functional demands, including feeding (*Fortey and Owens, 1999*) and burrowing behaviours (*Esteve et al., 2021*). Different trilobite clades display distinct suture patterns, with two sets, the circumocular and marginal sutures, relevant for the creation of this ecdysial gape (*Henningsmoen, 1975*). For the earliest diverging trilobites, olenellines (*Palmer and Repina, 1993*; *Lieberman, 2002*) which a have been recovered as a paraphyletic grade rather than a monophyletic group (e.g. *Paterson et al., 2019*) circumocular sutures facilitated shedding of the cornea during ecdysis (e.g. *Ramsköld and Edgecombe, 1991*) while a marginal suture (*Figure 1e*) facilitated anterior egression through separation of the lateral cephalic doublure from the dorsal cephalon (e.g. *Stubblefield, 1936*; *Henningsmoen, 1975*). In later diverging trilobites the circumocular and marginal sutures combined into a single system (*Figure 1d*). Here, the cornea is fused to the free cheek (e.g. *Stubblefield, 1936*; *Ramsköld and Edgecombe, 1991*) and these facial sutures – the fused circumocular and marginal sutures - facilitated both ecdysis of the visual surface and anterior egression of the exoskeleton following withdrawal from the free cheeks (e.g. *Henningsmoen, 1975*).

The origin of these fused dorsal ecdysial sutures – facial sutures – has traditionally been considered to fall within Trilobita (e.g. *Fortey and Whittington, 1989*; *Edgecombe and Ramsköld, 1999*; *Lieberman, 2002*). However, the presence of dorsal ecdysial sutures in earlier diverging artiopodans – the Protosutura (*Du et al., 2019*) and eye slits in Petalopleura (*Chen et al., 2019*) raised questions about the homology and origins within Artiopoda (e.g. *Hou et al., 2017b*; *Du et al., 2019*). An alternative hypothesis for the origins of the dorsal cephalic sutures emerged: that these had a deep root within Artiopoda and were subsequently lost in some groups including multiple times within Trilobita (*Hou et al., 2017b*; *Du et al., 2019*), rather than the traditional view that dorsal cephalic sutures in trilobites were derived within the clade, and thus these eye slits and facial sutures were acquired independently. Support for the deep root hypothesis comes from the variability of facial sutures in trilobites, and the recognition that convergence and loss of features is common when they have a function or allow adapation to a particular niche (e.g. *Moore and Willmer, 1997*). Further evidence for the variability of suture morphology comes from ontogenetic studies, such as the fusion of dalmanitinid facial sutures during ontogeny (*Drage et al., 2018*), and the loss of this feature in other trilobites such as '*Cedaria*' *woosteri* (*Hughes et al., 1997*). This indicates that there is scope for a facial suture to have been lost repeatedly in artiopodan groups earlier diverging than Trilobita, should this feature prove to have a deeper origin within Artiopoda (*Hou et al., 2017b*; *Du et al., 2019*). A deep root for facial sutures within Artiopoda would have repercussions for the importance of the facial sutures in determining trilobite relationships (e.g. *Jell, 2003*) and the position of a grade of olenelline trilobites as the earliest diverging members of the group (e.g. *Paterson et al., 2019*). To date, a third possibility – that the ventral plates of *Acanthomeridion* are homologous to the doublure of olenellids, and thus *Acanthomeridion* and olenelline trilobites share the presence of an unfused marginal suture – has not received

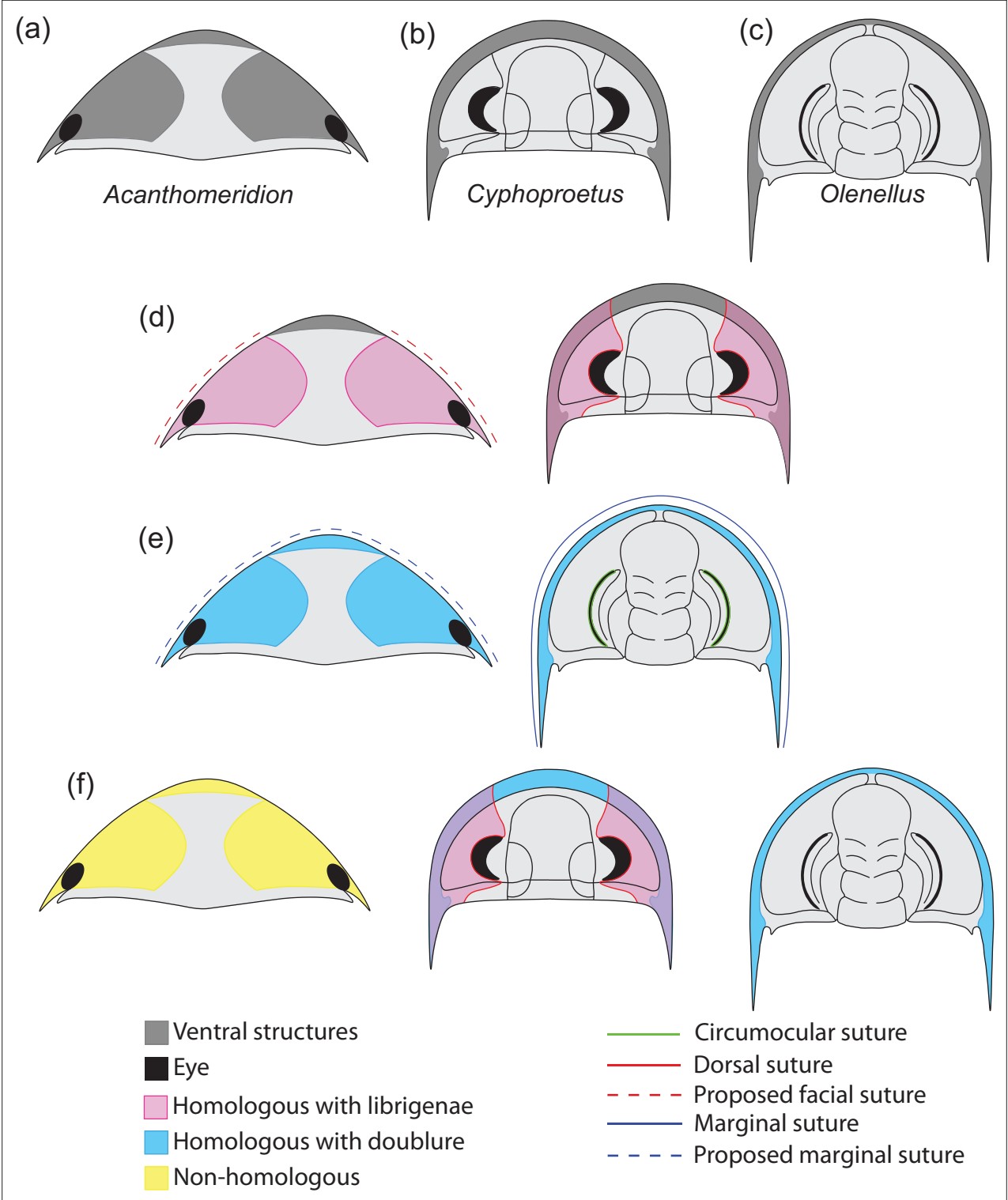

**Figure 1.** Hypotheses of homology in sutures between *Acanthomeridion* and trilobites. (**a**) Cephalic morphology of *Acanthomeridion serratum*. (**b**) Cephalic morphology of a proetid trilobite (modified from *Daley and Drage, 2016*). (**c**) Cephalic morphology of a redlichid trilobite (modified from *Whittington, 1989*). (**d**) Ventral plates and librigenae as homologous with dorsal suture. (**e**) Ventral plates and doublure as homologous with marginal suture. (**f**) Non-homology between ventral plates of *Acanthomeridion*, librigenae, and doublure.

broad attention in the literature, but should also be considered in light of the similarities in a position of the ventral plates and olenelline doublure, and the marginal rather than dorsal position of the suture in *Acanthomeridion*.

Resolving the complex history of dorsal ecdysial sutures evolution within Artiopoda, requires the nature of cephalic sutures in artiopodans such as *Acanthomeridion,* protosuturans, and petalopleurans to be resolved, and their phylogenetic position confidently determined (*Hou et al., 2017b*; *Du et al., 2019*). However, the morphology of many earlier diverging artiopodans is not completely understood (*Ortega-Hernández et al., 2013*; *Lerosey-Aubril et al., 2017*), contributing a significant amount of missing data to character matrices used in phylogenetic analyses. Up to now, *Acanthomeridion* was arguably the least well-known of these early diverging artiopodans, with its ventral anatomy and non-biomineralized structures very poorly known (*Hou and Bergström, 1997*; *Hou et al., 2017b*).

Here, we describe new specimens of *Acanthomeridion serratum* collected from Jiucun town, Chengjiang (*Figure 2*), using micro-CT to reveal details of the ventral anatomy, eyes, and appendages for the first time, and reconstruct the changing shape of the body during ontogeny. We use these new data to show that *A. anacanthus* is a junior synonym of *A. serratum*. We then assess the phylogenetic position of the species considering two hypotheses relating to the dorsal ecdysial sutures – as homologous to the dorsal ecdysial sutures (facial sutures combined of a fused marginal and circumocular suture) of trilobites (*Figure 1d*), that they are homologous to marginal sutures of olenellines (*Figure 1e*), and that these structures are unique to *Acanthomeridion* (*Figure 1f*), to evaluate the different hypothesis of origination within Artiopoda.

## Results

### Systematic palaeontology

ARTIOPODA Hou and Bergström, 1997

*Acanthomeridion* Hou et al., 1989

***Type and only species.*** *Acanthomeridion serratum* ***Hou et al., 1989***.

**Diagnosis.** Non-biomineralized artiopodan with an elongate dorsal exoskeleton that gives the body an elliptical outline. Subtriangular head shield with deep lateral notches that accommodate stalked elliptical eyes, rounded genal angles of the head shield, an axe-like hypostome, and paired teardrop-shaped plates on either side of the hypostome. Head bears large eyes and long multi-segmented antennae consisting of over 40 articles anterior to three pairs of small cephalic limbs. The trunk is composed of 11 tergites bearing expanded tergopleurae with well-developed distal spines, and a terminal spine. The ninth tergite bears a pleural spine more elongated than others, the eleventh tergite is reduced and expanded into a leaf-like outline. Each tergite bears a pair of biramous limbs with long and dense spines on the endopodites, and slender stick-like exopodites with long and dense bristles. Modified from *Hou et al., 1989*.

### *Acanthomeridion serratum* Hou et al., 1989 Syn. Nov.

*Figures 3–8*.

1989 *Acanthomeridion serratum* Hou et al., pl. III, figs. 1–5; pl. IV, figs. 1–5; p. 46, text-figs. 3, 4.

1996 *Acanthomeridion serratum* Chen et al., p. 157, figs. 199–203.

1997 *Acanthomeridion serratum* Hou and Bergström, p. 38.

1999 *Acanthomeridion serratum* Luo et al., p. 51, pl. 6 fig. 5.

1999 *Acanthomeridion serratum* Hou et al., p. 130, fig. 188.

2004 *Acanthomeridion serratum* Chen, p. 280, fig. 442.

2004 *Acanthomeridion serratum* Hou et al., p. 176, fig. 16.63; p. 177, fig. 16.64.

2017b *Acanthomeridion serratum*, *A. anacanthus*, and *Acanthomeridion* sp in Hou et al., p. 734, figs. 1–4.

2017a *Acanthomeridion serratum* Hou et al., p. 204, fig. 20.34; p. 205, fig. 20.35.

**Type material.** *Holotype*, CN 108305. *Paratype*, CN 108306–108310.

**New material**. Sixteen new specimens were collected from Jiucun (*Figure 2*), Chengjiang County, and housed in the Management Committee of the Chengjiang Fossil Site World Heritage (CJHMD 00052–00062), and Research Center of Paleobiology, Yuxi Normal University (YRCP 0016–0020).

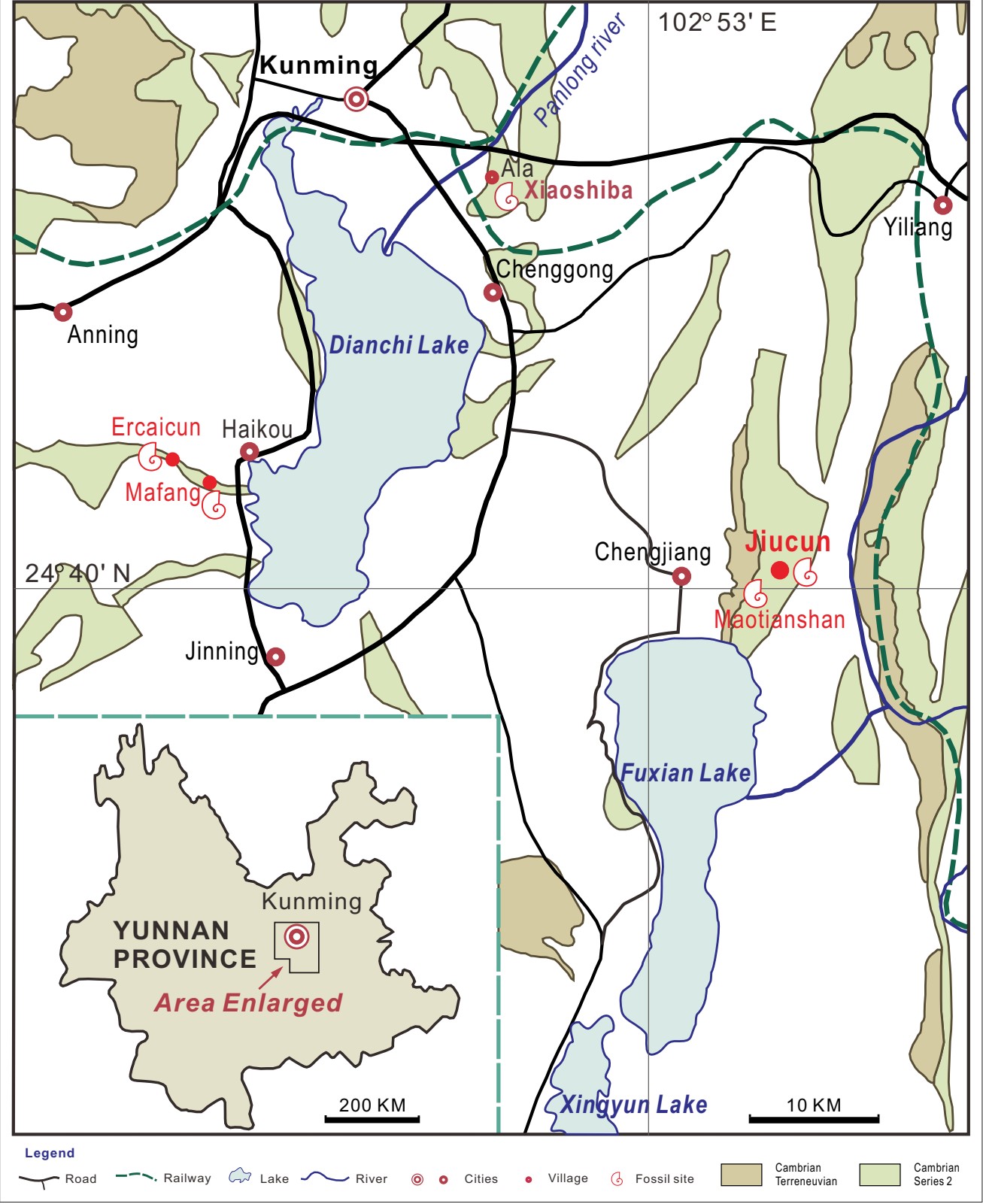

**Figure 2.** Sites yielding *Acanthomeridion* from the Cambrian Stage 3 Chengjiang Biota (red circles). All the specimens of *A. serratum* used here are collected from Jiucun town, Chengjiang County.

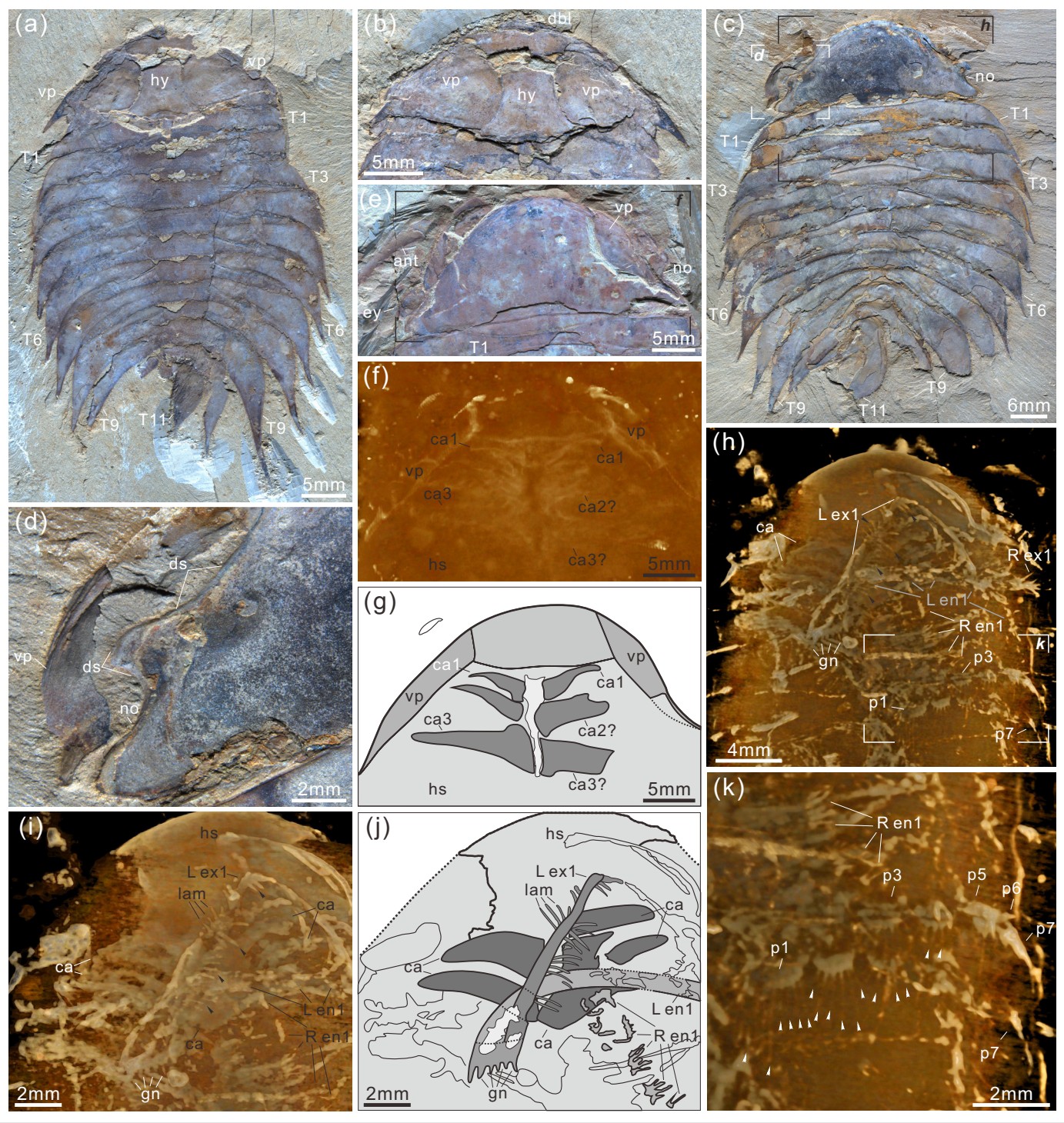

**Figure 3.** *Acanthomeridion serratum* from the Cambrian Stage 3 Chengjiang Biota. (**a, b**) CJHMD 00,052 a/b, respectively, an individual with hypostome, ventral plates, and 11 thoracic tergites. (**c, d**) CJHMD 00,053 a, showing ventral plate, dorsal sutures, and 11 thoracic tergites. (**e-g**) YRCP 0016 a, showing the ventral plates, and three post-antennal limbs. (**h-k**) Showing the post-antennal appendages under head, gnathobases of trunk limbs, stick-like exopodites with bristles (black arrows), and endopodites with long spines (white arrows). (**f**) Micro-CT image of YRCP 0016 a; (**h, i, k**) Micro-CT images of CJHMD 00,053 a. Abbreviations: ant, antenna; ca*n*, post-antennal appendage *n* beneath head; dbl, doublure; ds, dorsal suture; en, endopodites; ex, exopodites; ey, eye; hs, head shield; hy, hypostome; L, left; lam, lamellae; no, notch; p*n*, podomere *n*; R, right; T*n*, tergite *n*; ts, terminal spine; vp, ventral plate.

The online version of this article includes the following figure supplement(s) for figure 3:

*Figure 3 continued on next page*

*Figure 3 continued*

**Figure supplement 1.** *Acanthomeridion serratum* from the Cambrian Stage 3 Chengjiang Biota.

**Figure supplement 2.** *Acanthomeridion serratum* from the Cambrian Stage 3 Chengjiang Biota.

**Figure supplement 3.** *Acanthomeridion serratum* from the Cambrian Stage 3 Chengjiang Biota.

**Figure supplement 4.** *Acanthomeridion serratum* from the Cambrian Stage 3 Chengjiang Biota.

**Figure supplement 5.** *Acanthomeridion serratum* from the Cambrian Stage 3 Chengjiang Biota.

**Figure supplement 6.** *Acanthomeridion serratum* from the Cambrian Stage 3 Chengjiang Biota.

*Occurrence*. Yu'anshan Member, Qiongzhusi Formation, *Wutingaspis–Eoredlichia* biozone, Cambrian Stage 3. Jiucun and Maotianshan, Chengjiang County, Yunnan, China; Mafang and Ercaicun, Haikou town, Kunming, Yunnan, China (*Figure 2*).

*Diagnosis*. As for genus, by monotypy.

*Description*. The dorsal exoskeleton of *Acanthomeridion serratum* displays weak trilobation and elliptical outline. Specimens measure from 20 to 75 mm along the sagittal axis (*Figure 3a and c*; *Figure 4*; *Figure 5a*; *Figure 6*; *Figure 3—figure supplements 1–6*). The head shield is subtriangular in dorsal view, with a rounded anterior margin and small spines on the posterior margin (*Figure 3a and c*; *Figure 6*; *Figure 3—figure supplement 6*). A pair of lateral posterior notches accommodate stalked elliptical eyes. The head shield becomes wider relative to its length during ontogeny (*Figure 6*). The lateral margin of the head shield is a suture that attaches paired ventral plates. Ventral plates have a teardrop outline, occupy the entire length of the cephalon and terminate in a posterior spine (maximum length 3.4 mm; *Figure 3—figure supplement 2b*) that projects into the thorax as far as the anteriormost tergite (*Figure 4a*). The medial margin is curved towards the axial line and sits adjacent to a conterminant axe-shaped hypostome (maximum dimensions 9.6 mm × 11.6 mm; *Figure 3a and b*). Four pairs of appendages are present in the head (*Figure 3f–j*; *Figure 3—figure*

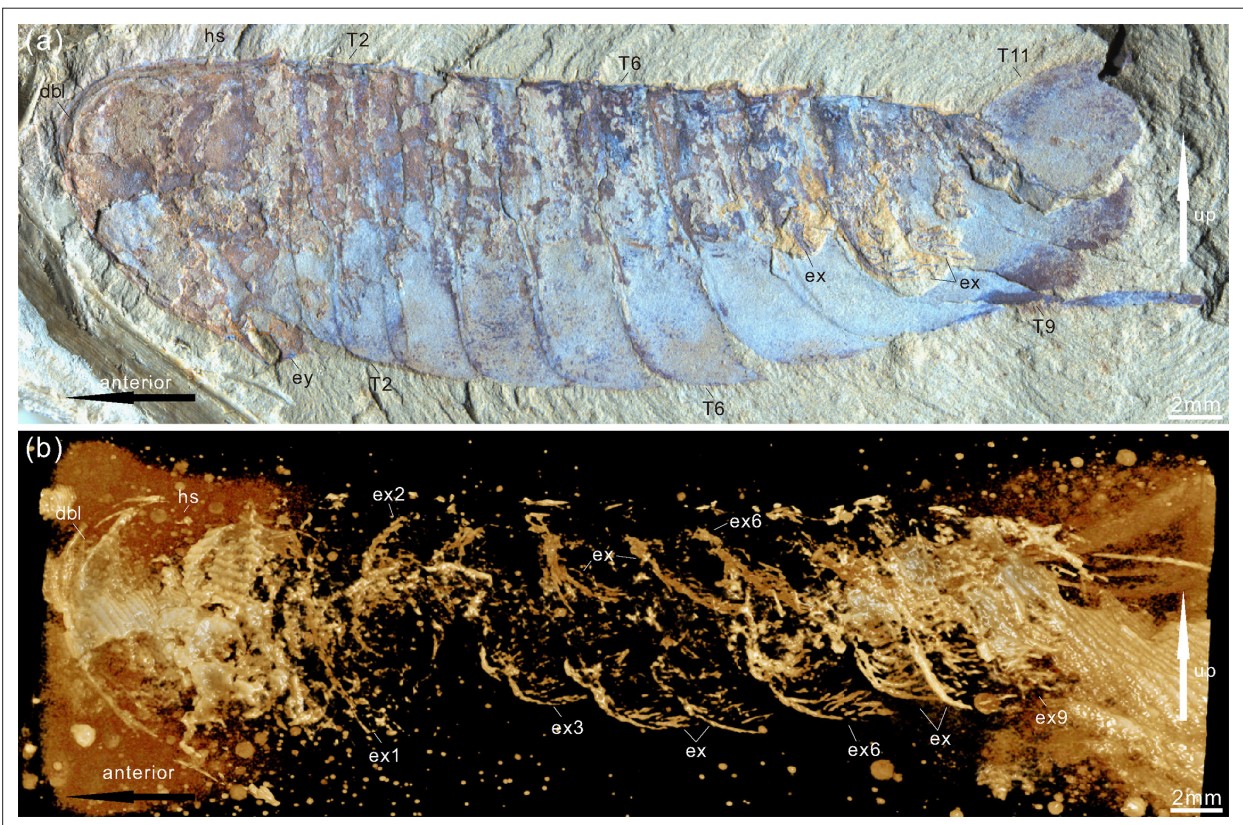

**Figure 4.** *Acanthomeridion serratum* from the Cambrian Stage 3 Chengjiang Biota. (**a**) CJHMD 00054, complete individual with head, eye, and 11 tergites. (**b**) Micro-CT image of panel (**a**) showing the stick-like exopods. Abbreviations: dbl, doublure; ex, exopod; ey, eye; es, eyestalk; hs, head shield; T*n*, tergite *n*.

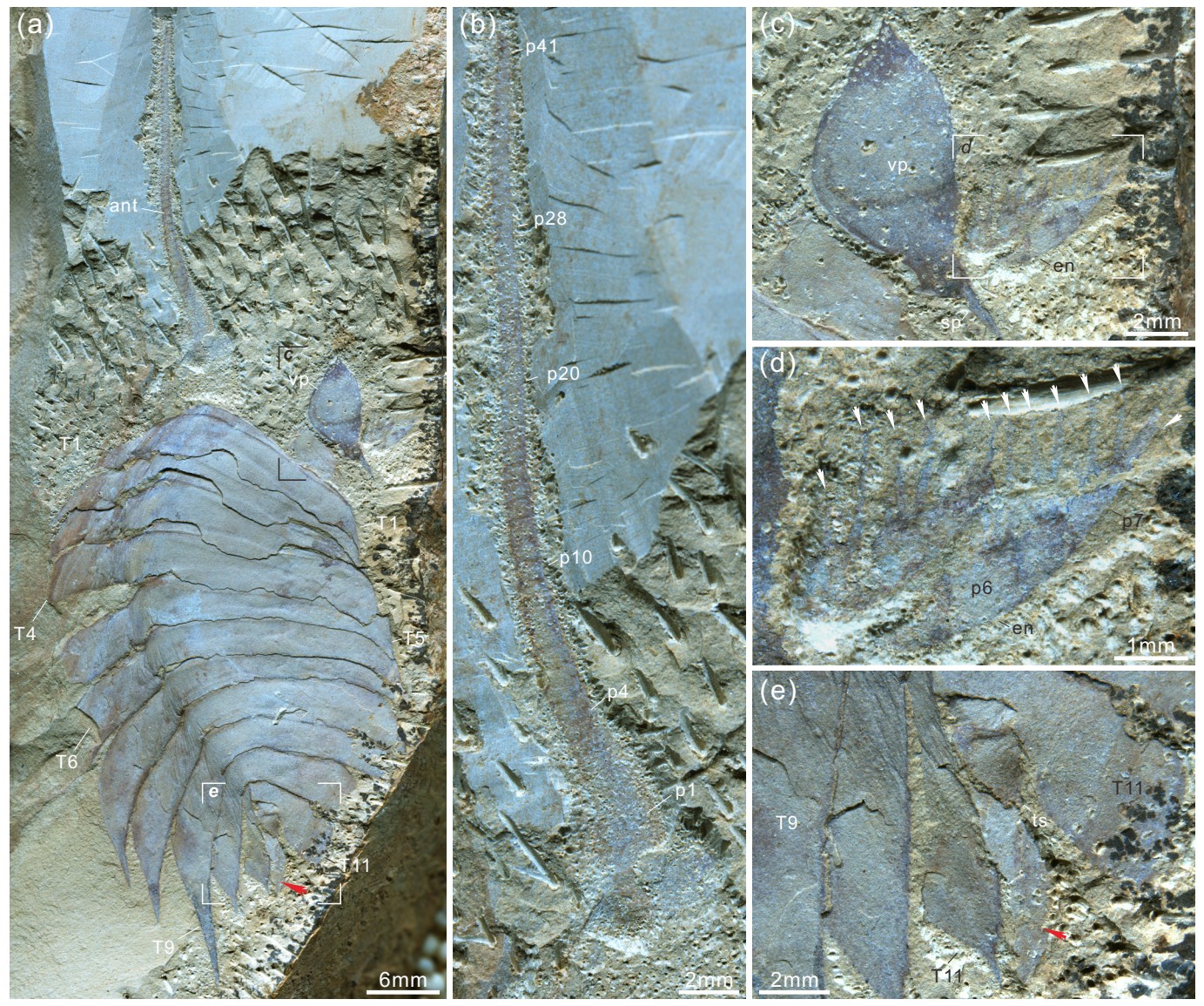

**Figure 5.** *Acanthomeridion serratum* from the Cambrian Stage 3 Chengjiang Biota. (**a–e**) CJHMD 00055, showing the antenna, ventral plate, endopodites with long spines (arrows in d), 11 tergites, and paddle-like structure (red arrow). (**a**) Overview of whole specimen. (**b**) Detail of long left antenna. (**c**) Close-up of ventral plate. (**d**) Details of long spines (arrows) of right endopodites. (**e**) Close-up of paddle-like structure (red arrow). Abbreviations same as *Figures 3 and 4*.

supplement 5a). A pair of antennae with at least 43 articles (*Figure 3—figure supplement 2a, f; Figure 3—figure supplement 4; Figure 5a and b*) sits anterior to three pairs of small cephalic limbs (*Figure 3f–j*). Articles of the antennae decrease in length and width distally. The preserved part of the antennae is slightly shorter than the trunk (*Figure 5*). The cephalic appendages are poorly preserved and only the gross morphology can be discerned as no individual podomeres are visible. The protopodite of the cephalic limbs are subtriangular in outline. Endopodites and exopodites are not clearly preserved. Appendage three may include the endopodite, but no podomeres are visible.

The trunk is composed of 11 tergites which extend laterally into spinose tergopleurae. Tergites are all subequal in length. Tergites 8–11 curve towards the posterior, and reduce in width progressively, so that T11 is approximately 25% the width of T8 and 33% the width of T9 (*Figure 3a and c; Figure 4, Figure 5a; Figure 3—figure supplement 1*). Each tergite has small spines on its lateral and posterior margins (*Figure 4a; Figure 7; Figure 3—figure supplement 1; Figure 3—figure supplement 2i, k; Figure 3—figure supplement 3a, j; Figure 3—figure supplements 5b, e, f and 6a–f*).

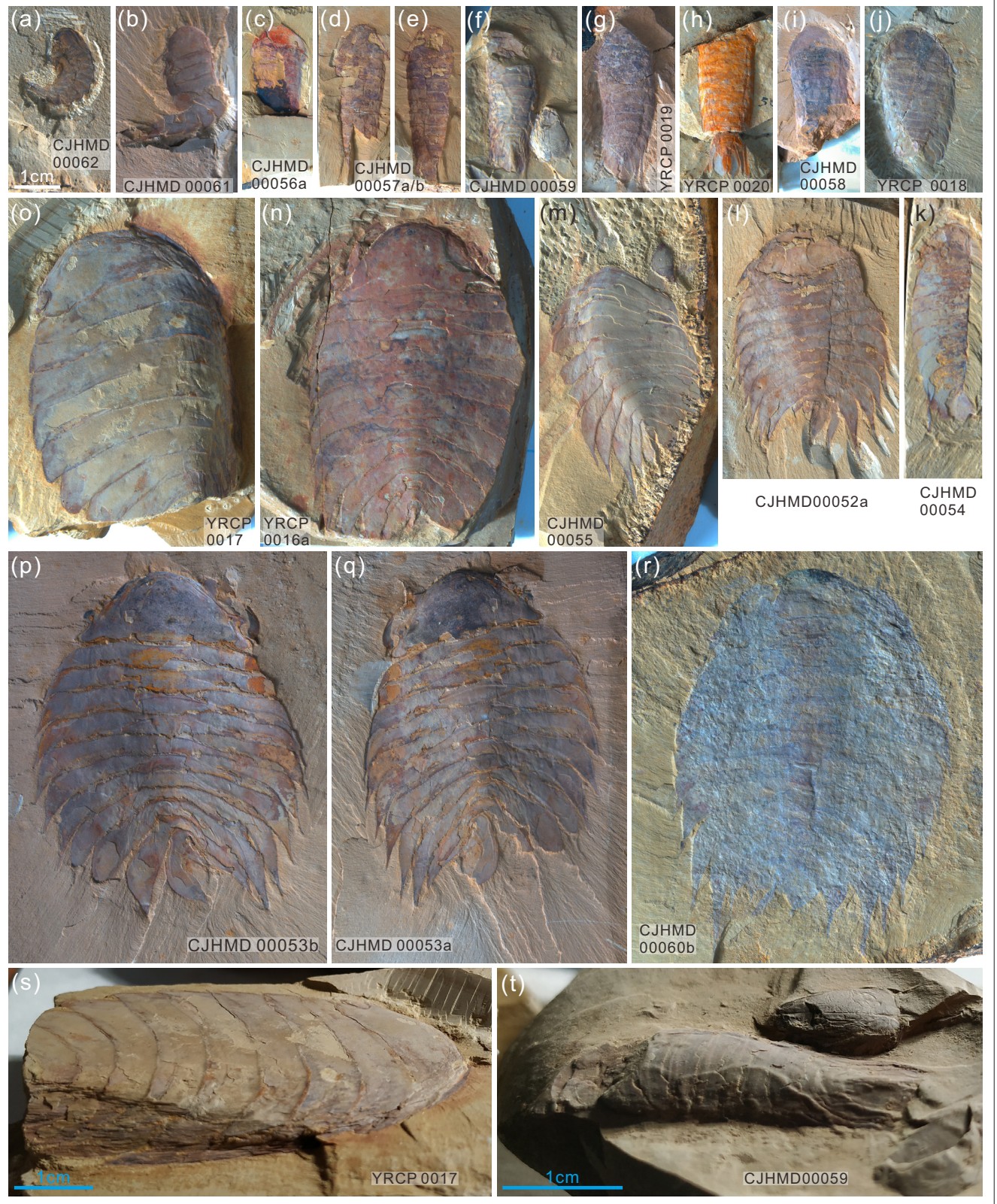

**Figure 6.** Ontogenetic series (**a–r**) of *Acanthomeridion* and their ventrally curling pleurae (**s, t**). (**a–r**) Showing the individuals from smallest to largest with same scale bar. (**s**) Lateral view of (**o**), note the right curling pleurae and left flat pleurae. (**t**) Lateral view of (**f**), showing the left curling pleurae.

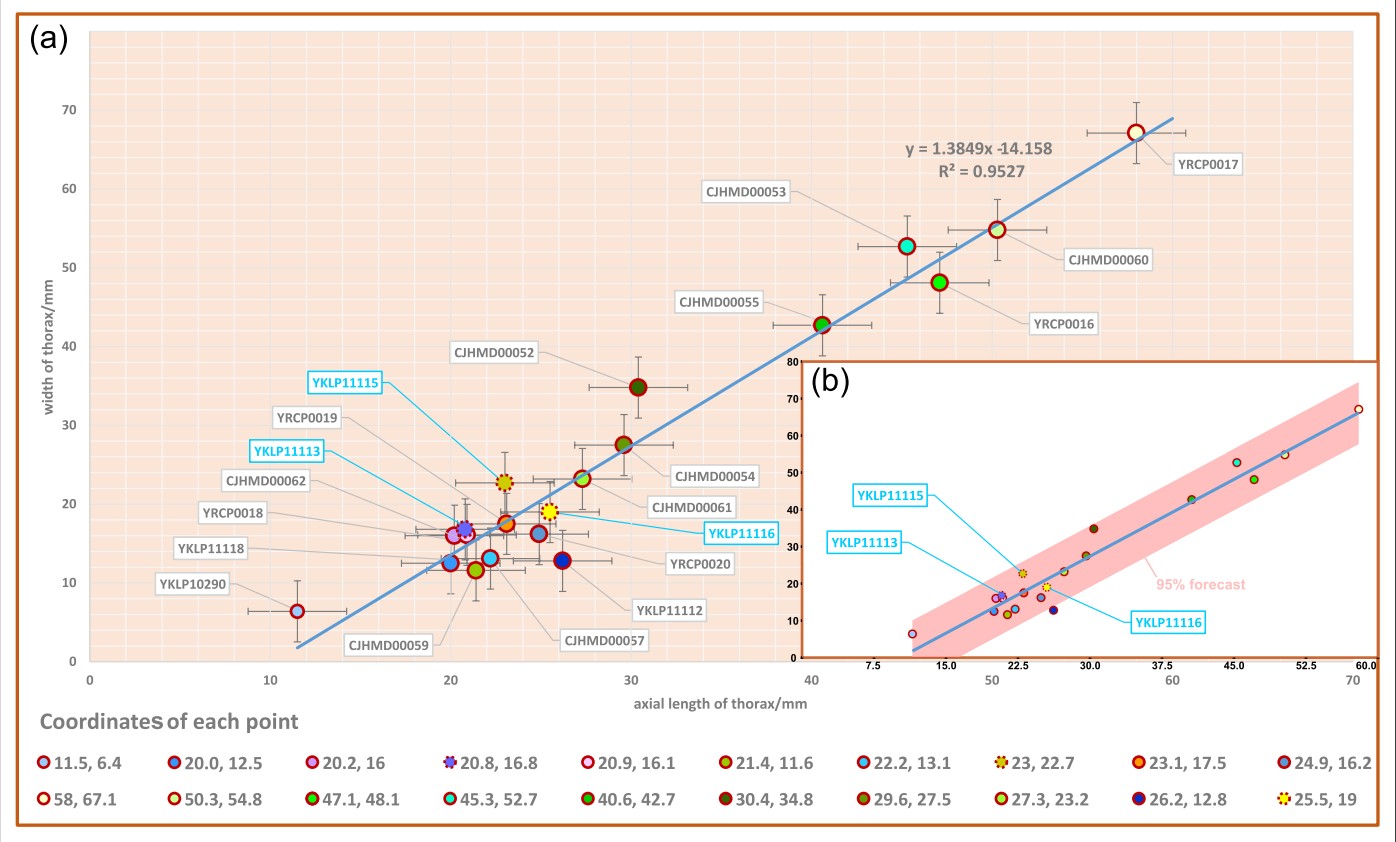

**Figure 7.** Scatterplot of axial length and width of thorax of *Acanthomeridion serratum*. (**a**) Scatterplot of raw data. (**b**) Data with a 95% forecast shown in pink. Specimens previously assigned to *A. anacanthus* indicated by blue lettering.

The shape of tergites and the convexity of the trunk changes through ontogeny (*Figure 7*). In the smaller specimens, the trunk appears slender from a dorsal view as the body displays a high convexity (*Figure 6t*). Pleurae curve ventrally, and only slightly towards the posterior (*Figure 6a–k*). In larger specimens, the trunk is less convex than for the smaller specimens (*Figure 6s*), ventral curvature of pleurae is decreased, and posterior curvature is increased (*Figure 6o–r*). A linear relationship between trunk axial length and width is obtained (*Figure 7*). Where trunk appendages are preserved, each tergite is associated with a pair of biramous trunk appendages (*Figure 3a, c, h and k*; *Figures 4 and 5*). The protopodite is sub-triangular with a broad attachment to the body wall and short robust gnathobasal spines along the medial margin (*Figure 3h–j*; *Figure 3—figure supplement 3a–f, i*). The ventral margin of the protopodite has endites longer than the gnathobasal spines. The dorsal edge of the protopodite is poorly preserved and does not clearly show the attachment of the exopodite. Exopodites are slender, stick-like, and long (8.3 mm, ca. 3 x protopodite length), bearing two rows of lamellae (*Figure 3h, i and k*; *Figure 4*). Exopodites are shorter than the width of corresponding tergites (*Figure 3h*; *Figure 4*). The lamellae appear short but may not be completely preserved. There is no evidence that the exopodite is divided into lobes or has articulations. Endopodites are composed of seven trapezoidal podomeres. Podomeres 1–6 bear spiniferous endites arranged in rows, six on pd1, 2, 4 on pd3, 4, and 3 on pd5, 6. Podomere seven displays a terminal claw of three elongated spines (*Figure 3h and k*; *Figure 5d*). From the preserved appendages, no significant anteroposterior differentiation or dramatic size reduction can be discerned (*Figures 3 and 4*). One specimen shows evidence of paired midgut diverticulae (*Figure 3—figure supplement 4a, f*). The body terminates in a long, slender spine, dorsal to a paired paddle-like structure (*Figure 5a and e*; *Figure 3—figure supplement 5b, g*; *Figure 3—figure supplement 6g, i*).

**Remarks**. The eyes were previously identified as genal projections (see Figure 1, Figure 2A, and Figure 3A in *Hou et al., 2017b*) because of the limited available specimens and the posterior location

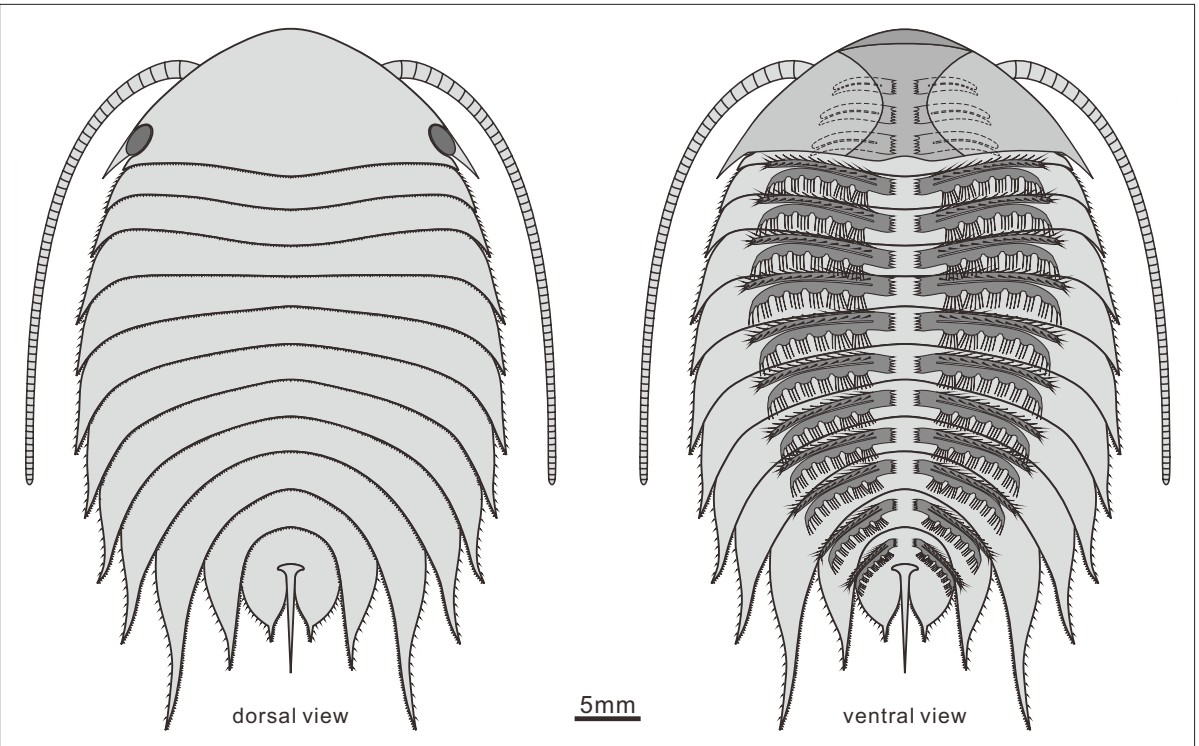

**Figure 8.** Reconstruction of *Acanthomeridion serratum* in dorsal and ventral view. Only the protopodite of the head appendages has been observed. There appear to be three post-antennal limbs with the fourth pair below the cephalic-thoracic boundary.

of the eyes. *Acanthomeridion anacanthus Hou et al., 2017b* is here proposed to be a junior subjective synonym of *Acanthomeridion serratum Hou et al., 1989*, an interpretation supported by:

1. The paddle-like structure under the terminal spine (*Figure 5a and e*; *Figure 3—figure supplement 5b, g*) was previously interpreted as the twelfth tergite. The specimen YKLP 11115 (see Figure 4 in *Hou et al., 2017b*) which is a similar size to YKLP 11116 (holotype of *Acanthomeridion anacanthus*) and YKLP 11113 (see Figure 2A, C and Figure 3A, C in *Hou et al., 2017b*) show 11 tergites.
2. Specimens CJHMD 00052 to –55 and YRCP 0016 show the evidently pleural spines of T8 and T9 (*Figure 3a and c*; *Figure 4a*; *Figure 5a*; *Figure 3—figure supplement 1*; *Figure 3—figure supplements 2a and 4b, g*).
3. The orientation of pleurae makes some specimens appear narrower than others. Specifically, those where pleurae curve ventrally look more slender than those where pleurae are curved posteriorly. Specimen YRCP 0017 preserves the right ventrally curving pleura and left flat pleura, which leads to its right pleura being narrower than the left (*Figure 6j, o and s*; *Figure 3—figure supplement 2i*). Moreover, most small specimens (1cm–3cm) are preserved with ventrally curving pleurae (*Figure 6a–i*; *Figure 3—figure supplement 3a–c*; *Figure 3—figure supplement 4a*; *Figure 3—figure supplement 5a–c*; *Figure 3—figure supplement 6g, i*) and these appear narrower than the few small specimens with flat pleurae (*Figure 6j*; *Figure 3—figure supplement 4b*. See also Figure 2A–D, and Figure 4 in *Hou et al., 2017b*). Larger specimens are typically preserved with flattened, posteriorly curving pleurae.
4. Twenty specimens show a good ontogenetic series of *Acanthomeridion* including some that have previously have been assigned to '*Acanthomeridion anacanthus*' (*Figures 6 and 7*). Linear measurements do not support distinct species of different sizes, nor with different proportions of thorax axial length to width. The length and width data fit well with a linear equation; most of the data fall within the 95% forecast (*Figure 7b*). Data fall close to the fitted line, as indicated by the $R^2$ value of 0.95. Lastly, specimens previously assigned to '*A. anacanthus*' are recovered as smaller *A. serratum* specimens with similar thoracic shapes.

The new specimens and CT data presented herein allow the description of cephalic and trunk limbs, and a conterminant hypostome adjacent to ventral plates in *Acanthomeridion* for the first time

(reconstruction in *Figure 8*). The presence of a terminal spine on these ventral plates is identified. The ventral plate is separated from the dorsal part of the cephalon by a suture that runs close to the stalked eyes. The possible affinities of these ventral plates, and the phylogenetic implications of these, are discussed below (see Discussion). The presence of four appendages in the head (or three in the head and one underneath the articulation) is comparable to *Zhiwenia* (*Du et al., 2019*), as well as other artiopodans including trilobites (*Edgecombe and Ramsköld, 1999*; *El Albani et al., 2024*). However, the morphology of the three non-antenniform cephalic limbs is difficult to determine with only the protopodite clearly visible. The lack of observed endopodites and exopodites on these three appendages may be a preservational artefact. Possibly the absence of clear endopodites and exopodites represents specialization of the cephalic appendages with the structures being significantly reduced. Other Cambrian artiopodans have been shown to have appendage specialization (*Losso and Ortega-Hernández, 2022*) which is frequently found in the cephalon (*Chen et al., 2019*; *Schmidt et al., 2022*), but none show a reduction of both the exopodite or endopodite. In *Sinoburius lunaris*, the first two post-antennal appendages have reduced endopodites with elongated exopodites (*Chen et al., 2019*), and in *Pygmaclypeatus daziensis* the exopodite is significantly reduced in the cephalic appendages (*Schmidt et al., 2022*).

Recently, the head of *Retifacies abnormalis* was interpreted to develop three pairs of uniramous post-antennal appendages and followed by one biramous appendage pair (*Zhang et al., 2022*). It would be unclear the function of an appendage with both distal elements being significantly reduced until more clear appendages are found.

The exopodites of *Acanthomeridion* also differ substantially from those known in other artiopodans. Specifically, the stick-like nature of the exopodite, the rows of lamellae combined with the apparent lack of segmentation, differ from the paddle-shaped exopodites thought to characterise the remainder of the group (e.g. *Schmidt et al., 2022* and Figure 4 in *Ortega-Hernández et al., 2013*).

## Possible affinities of the ventral plates in *Acanthomeridion*

### Ventral plates as homologs of trilobite free cheeks

The ventral plates of *Acanthomeridion* have previously been suggested to be homologs to the free cheeks of trilobites, with the suture between these features and the remainder of the cephalon interpreted as a facial suture (*Figure 1d*) (e.g. *Hou et al., 2017b*). Morphological similarities are apparent between the ventral plates and free cheeks in many trilobites (*Figure 9a and b*; *Whittington et al., 1997*) as well as the path of suture separating them from the rest of the cephalon. Our new data provide additional support for interpreting the ventral plates as homologus of the librigenae. The ventral plates of *Acanthomeridion* are here shown to be transversely broad, tapering posteriorly to a spine, in a position directly comparable to the genal spine of many trilobites (*Figure 1d*; *Figure 9a, b*). Furthermore, the ventral plate is separated from the rest of the cephalon by a suture that passes near the compound eye. This path is similar to the facial suture (fused circumocular and marginal sutures) of many trilobites especially those bearing opisthoparian facial sutures, which separate the free cheeks from the cranidium passing through the visual surface (*Figure 9a and b*; *Whittington et al., 1997*). The homology of eye notches (e.g. *Luohuilinella* and *Zhiwenia* [*Figure 9e and f*]), and eye slits (e.g. *Phytophilaspis Ivantsov, 1999*; *Figure 9c and d*) of petalopleurans, dorsal ecdysial sutures in trilobites and *Acanthomeridion* has been suggested previously (*Du et al., 2019*). These eye slits join the dorsal eye with the anterolateral margin of the head shield, possibly forming a single suture that facilitated the release of the visual surface and created an anterior gape for the animal to leave the old exoskeleton during ecdysis. However, the evolution from eye notches to eye slits and finally to the loss of the eye slit has been inferred, calling into question the affinity between these structures and sutures (*Chen et al., 2019*). This hypothesis was tested explicitly in the first matrix, where the ventral plates of *Acanthomeridion* were considered homologous to the free cheeks of trilobites. In this analysis, the dorsal slits and sutures of other artiopodans were also treated as homologous ecdysial sutures in the cephalon (*Figure 10a*).

### Ventral plates as homologs to doublure in olenelloid trilobites

In the above scenario, the ventral plates of *Acanthomeridion* would be homologous to the free cheeks both dorsally and ventrally in trilobites, including the doublure. However, it is possible that the ventral plates are only homologous to the doublure, and thus that the suture represents the marginal suture,

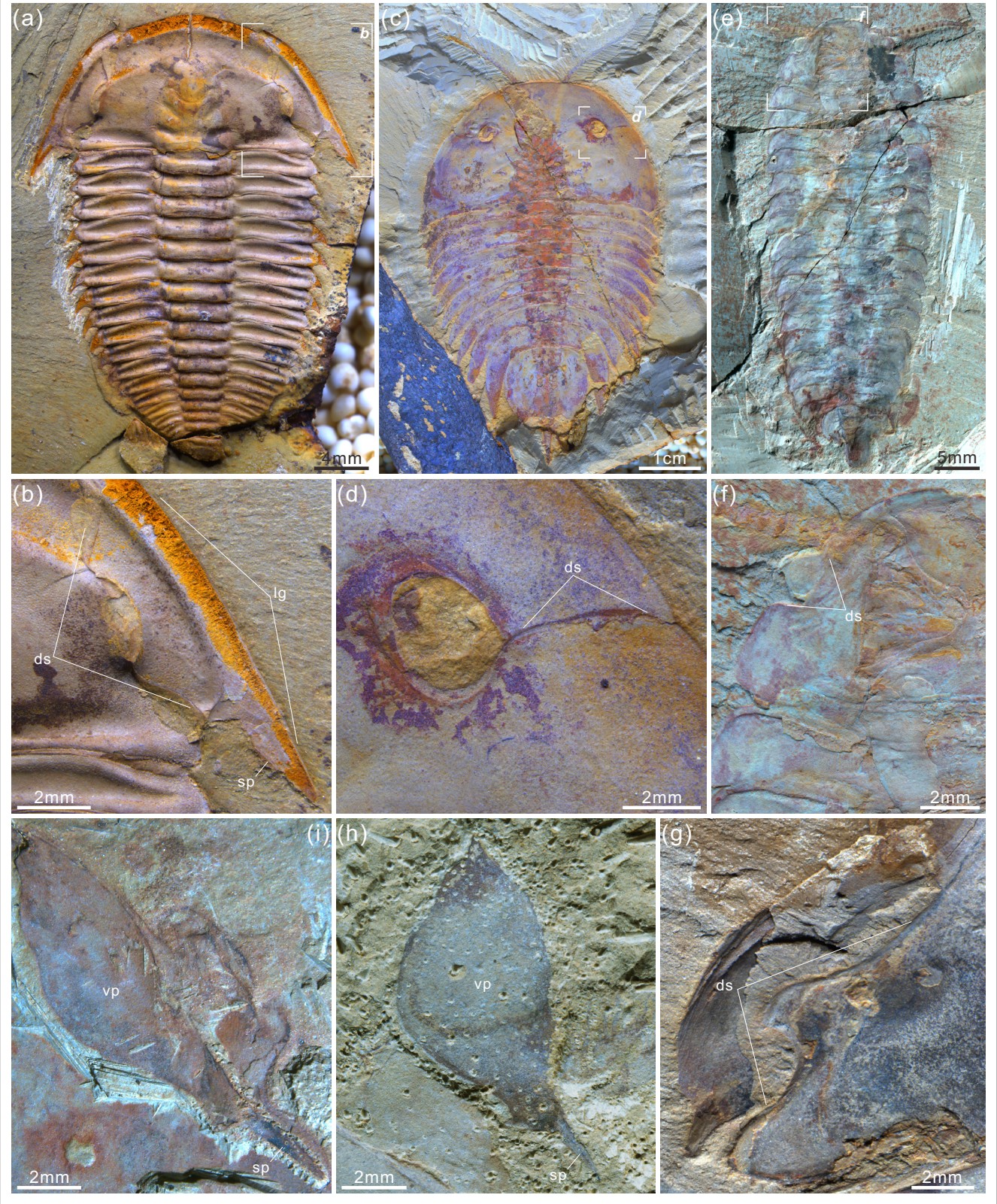

**Figure 9.** Four representative artiopodans from early Cambrian with their dorsal sutures, free cheeks, and ventral plates. (**a, b**) Trilobite *Wutingaspis tingi* from Chengjiang Biota, note its free cheeks and dorsal or facial sutures. (**c, d**) Xandarellid artiopodan *Xandarella spectaculum* from Chengjiang Biota bearing the distinctive dorsal sutures. (**e, f**) Holotype of the protosuturan artiopodan *Zhiwenia coronata* from Xiaoshiba Biota developing dorsal sutures. (**g**) The left dorsal suture of *Acanthomeridion serratum* from Chengjiang Biota, showing the morphological and positional similarities to that

*Figure 9 continued on next page*

*Figure 9 continued*

of *W. tingi* (**a, b**), *X. spectaculum* (**c, d**), and *Z. coronata* (**e, f**). (**h, i**) Right ventral plates of *A. serratum* from Chengjiang Biota bearing a terminal spine, which is similar to free cheek of trilobite like *W. tingi* (**a, b**).

rather than a fused facial suture. The difficulty lies in the marginal position of the eyes, which could support the interpretation that these are fused sutures but might also represent a position that facilitated ecdysis through a single, unfused, suture. Considered homologous only to the doublure, the suture in *Acanthomeridion* separating the ventral plates from the cephalon would be homologous to the unfused, marginal suture in olenelloids rather than the fused facial suture of most other trilobites (*Figure 1e*) (e.g. *Stubblefield, 1936*).

Few morphological features can be drawn to support this hypothesis beyond the ventral position of the plates. In particular, the lateral position of the eyes in *Acanthomeridion*, and the path of the suture close to these eyes, suggests that the suture functioned to release both the ventral plate and the eye. Although some artiopodans are known with expansive doublure such as *Squamacula buckorum* (*Paterson et al., 2012*) this structure usually follows the outline of the margin and lacks spines (*Whittington et al., 1997*).

This hypothesis was tested in the second matrix, where the ecdysial suture of *Acanthomeridion* was considered unfused (*Figure 10b*). In this scenario, the presence of eye slits and other dorsal sutures in other artiopodans would not represent homologous structures, as the proposed evolution is from a ventral-lateral structure to a dorsal one. Thus, the eye slits of the petalopleurans who bear them were not considered as ecdysial sutures for this analysis.

## Ventral plates and eye slits of artiopodans as unique structures without homologs in trilobites

Eye slits of petalopleuran artiopodans as homologus to facial sutures of trilobites have drawn support from comparisons such as *Loganopeltoides* (*Rasetti, 1948*), where a suture charts a path from the dorsal eye to the anterolateral margin of the head shield. However, in *Loganopeltoides* this suture represents the vestiges of the free cheeks (*Edgecombe and Ramsköld, 1999*), a derived state within Trilobita. Given that the features in support of homology between the ventral plates of *Acanthomeridion* and trilobite-free cheeks, and implicit within this a deep root of the facial suture within Artiopoda rather than Trilobita, may not represent true homologus (e.g. spines are often evolved convergently, and the dorsal eye slits of artiopodans have previously been considered unique characters – see e.g. *Edgecombe and Ramsköld, 1999*; *Chen et al., 2019*) it is important to consider the hypothesis that these features are unique to *Acanthomeridion*. Thus for this third analysis, the ventral plates of *Acanthomeridion* were treated as their own character in the matrix (*Figure 10c*).

### Phylogenetic analyses

All six Bayesian analyses produced the same broad relationships within Artiopoda, with the exception of the position of *Acanthomeridion* (*Figure 10*). Petalopleurans were recovered sister to Nektaspida, *Australimicola* sister to Conciliterga, and Vicissicaudata was not completely resolved. The positions of *Bailongia, Emeraldella, Kwanyinaspis, Squamacula,* and *Zhiwenia* were not resolved, and the internal relationships of Artiopodans were also not well resolved. *Acanthomeridion* was recovered as a sister to Trilobita when the ventral plates were considered homologous to the free cheeks of trilobites, but otherwise, its position was not well resolved. Within Trilobita, when the analysis was unconstrained, *Olenellus* and *Eoredlichia* formed a group sister to the other trilobites, or these two taxa were recovered in a polytomy with a clade containing the other trilobites (*Figure 10—figure supplements 1–3*). When the group *Anacherirurus, Cryptolithus, Eoredlichia, Olenoides,* and *Triarthrus* was constrained (with the exclusion of *Olenellus*), then *Olenellus* was recovered as sister to all other trilobites (*Figure 10—figure supplements 1–3*). This result is important for considering the hypothesis that the ventral plates of *Acanthomeridion* were homologous to the doublure of trilobites.

When visualized in treespace, constrained and unconstrained analyses produce very similar results (*Figure 11*). Trees in the posterior sample of the analysis where the ventral plates of *Acanthomeridion* were treated as homologous to the free cheeks of trilobites fall in the area of treespace supporting *Acanthomeridion* as sister to trilobites. This result is corroborated by the raw counts in the full sample

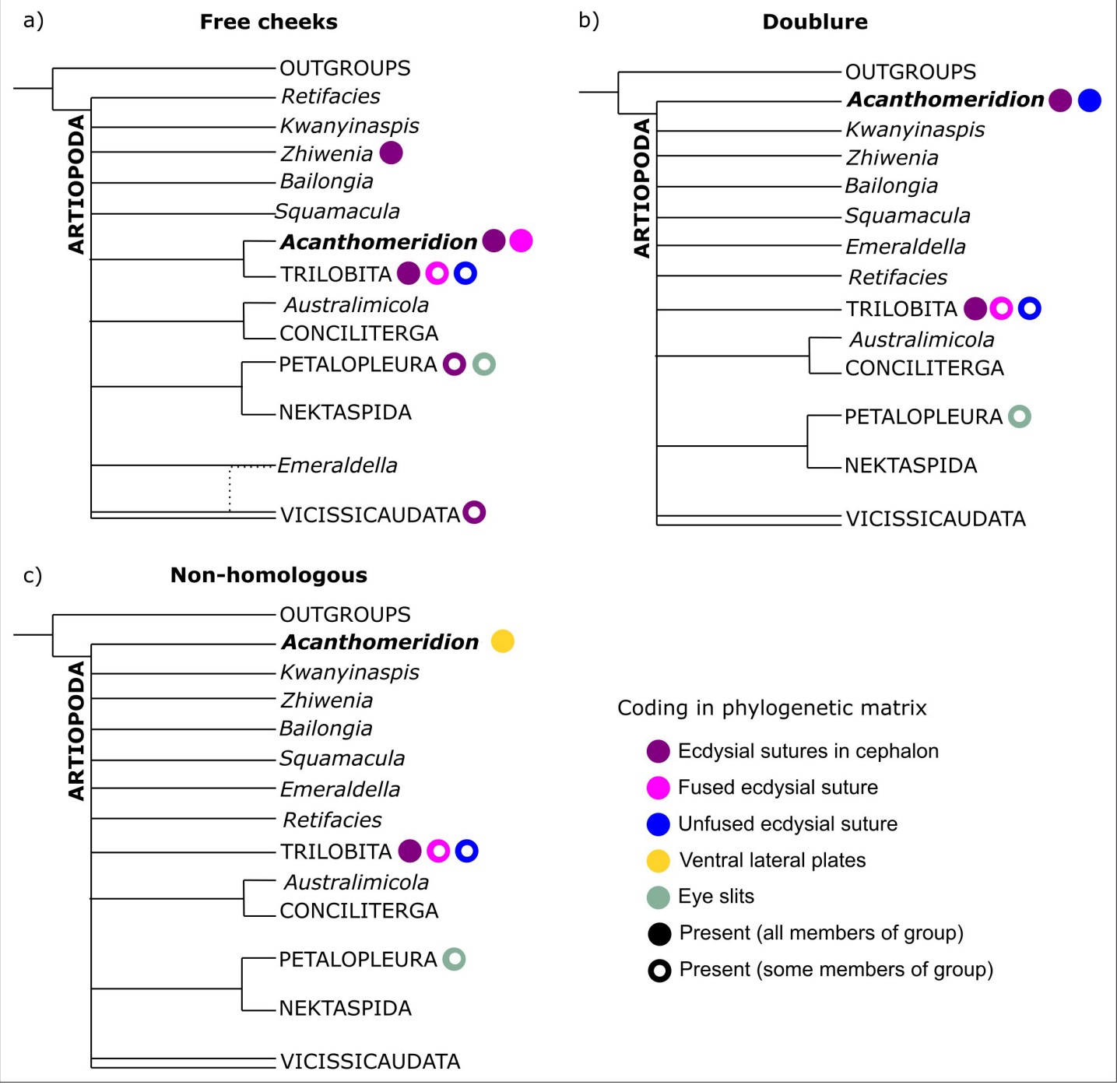

**Figure 10.** Simplified results of phylogenetic analyses. Comparison of results from matrices with different coding strategies, with coding of key characters for terminals within the analysis indicated by colored circles. (**a**) Ventral plates are considered homologous to free cheeks. (**b**) Ventral plates are considered homologous to trilobite doublure. (**c**) Ventral plates are not considered homologous to any artiopodan character.

The online version of this article includes the following figure supplement(s) for figure 10:

**Figure supplement 1.** Results of phylogenetic analyses using Bayesian inference, with the ventral plates of *Acanthomeridion* treated as homologous to the librigenae of trilobites.

**Figure supplement 2.** Results of phylogenetic analyses using Bayesian inference, with the ventral plates of *Acanthomeridion* treated as homologous to cephalic doublure of trilobites.

**Figure supplement 3.** Results of phylogenetic analyses using Bayesian inference, with the ventral plates of *Acanthomeridion* treated as non homologous to any cephalic feature of trilobites.

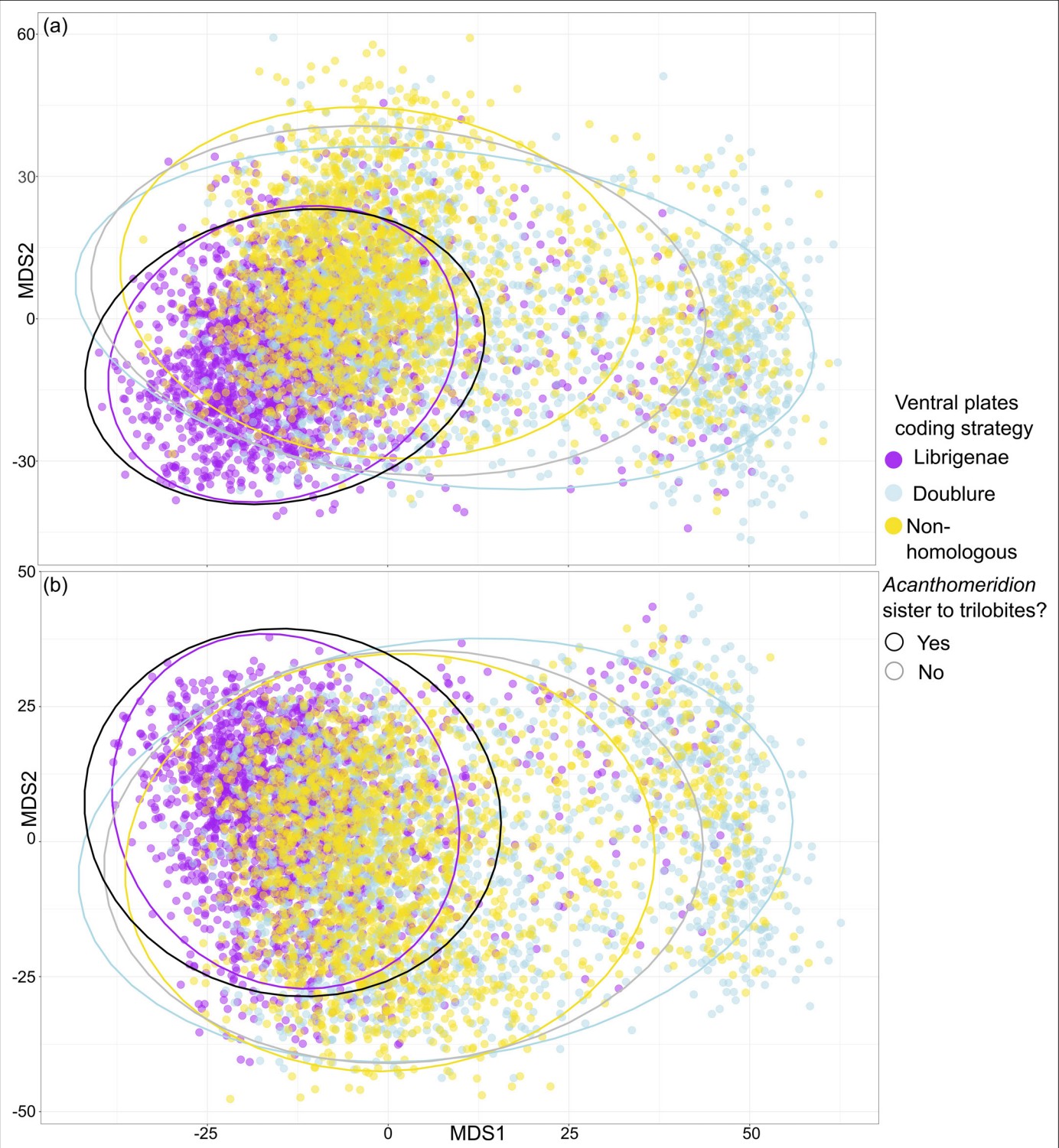

**Figure 11.** Treespace visualization of Bayesian Inference phylogenetic analyses. Multidimensional treespace plotted by bipartition relating to coding strategies (color of point). (**a**) Analyses were unconstrained. (**b**) Trilobites were constrained so that *Olenellus getzi* was resolved as the earliest diverging member of the group. Area included within a 95% confidence interval (CI) indicated by points inside the line of different colors. Area where *Acanthomeridion* is recovered sister to trilobites (95% CI) also shown by area within black line. Where *Acanthomeridion* is not recovered sister to trilobites (95% CI), shown by area within gray line. Note the overlap in the areas of the black and purple lines.

**Table 1.** Number of trees in posterior sample supporting *Acanthomeridion* as sister to trilobites / number of trees in the posterior sample total.

Results tabulated for each coding strategy, with unconstrained and constrained analyses (constrained analyses where a monophyletic group of all trilobites except for *Olenellus getzi* was forced, in order to recover trilobite relationships compatible with *Paterson et al., 2019*).

|  | Unconstrained | Constrained |
| --- | --- | --- |
| Homologous to free cheeks | 7832/10504 (75%) | 8083/10504 (77%) |
| Homologous to doublure | 1496/12004 (12%) | 1836/10504 (17%) |
| Not homologous | 761/12004 (6%) | 926/10504 (9%) |

(*Table 1*), and by the support for this node as seen in the consensus tree (*Figure 11*). Posterior samples of analyses treating ventral plates as homologous to the doublure and those treating them as non-homologous with any trilobite feature overlapped in treespace, over a much broader region. These display a very similar confidence ellipse to the sample of trees where *Acanthomeridion* is not resolved as sister to trilobites, but instead in a different position in the tree (*Figure 11*). Again, these treespace results are corroborated by the raw counts in the full sample (*Table 1*) and by the poorly resolved position of *Acanthomeridion* in the consensus trees (*Figure 11*).

## Discussion

All consensus trees support multiple origins of cephalic sutures in Artiopoda (*Figure 10*, *Figure 10—figure supplements 1–3*). When the sutures of *Acanthomeridion* and trilobites are considered as homologous (e.g. *Hou et al., 2017a*), as well as to those of other artiopodans (e.g. *Du et al., 2019*), terminals with ecdysial sutures in the cephalon are found in multiple groups in Artiopoda (*Figure 10a*), although *Acanthomeridion* is recovered sister to trilobites. If instead the sutures in *Acanthomeridion* are considered marginal, and homologous to those of olenelline trilobites, *Acanthomeridion*, and trilobites do not form a clade (*Figure 10b*, *Table 1*), and trees occupy a different region of the treespace (*Figure 11*). Indeed, the posterior samples of analyses treating the ventral plates as not homologous to any other artiopodan feature (*Figure 10c*) are extremely similar to those treating them as homologous to the doublure of trilobites, as illustrated by their broad overlap in treespace (*Figure 11*) and similar support for nodes in the consensus trees (*Figure 10—figure supplements 1–3*). Thus, two of the three coding strategies receive some support from the phylogenetic results. Either *Acanthomeridion* shares the presence of facial sutures with trilobites, and these form a clade, but the sutures in petalopleurans and *Zhiwenia* are distinct (multiple origins of dorsal ecdysial sutures) or the ventral lateral plates of *Acanthomeridion* are a distinct feature to the free cheeks of trilobites (and petalopeluran and *Zhiwenia* features are also distinct; multiple origins of sutures). The position of *Acanthomeridion* as not sister to trilobites in the analyses treating the ventral plates as homologous to doublure means that this hypothesis did not gain support, given the assumptions and character matrices of this study.

Support for *Acanthomeridion* as sister to trilobites comes from morphological similarity between the spinose termination of the ventral plates of *Acanthomeridion* (similar to trilobite genal spines) and the suture that follows an oblique line and passes near the eye both for *Acanthomeridion* (similar to trilobite opisthoparian facial sutures; *Whittington et al., 1997*). Taken together with trilobate exoskeleton of large *Acanthomeridion* specimens, the non-biomineralised *Acanthomeridion* bears many morphological similarities to biomineralised trilobites that further support a sister relationship. However, this sister relationship rests on the treatment of the ventral plates as free cheeks. If this interpretation is not favored, then these similarities can be considered convergent, and the position of *Acanthomeridion* within Artiopoda is not well resolved by these analyses. For this interpretation of the ventral plates, the path of the suture between the ventral plates and cephalon passing close to the eye may represent a way to minimise damage to the eye during ecdysis, which is comparable to what is observed in trilobites with facial sutures.

Other features observed in *Acanthomeridion* for the first time – uniramous deutocerebral appendages, an axe-shaped conterminant hypostome, two librigena-like plates, and three additional head

appendages – are consistent with a phylogenetic position for the species as an early diverging artiopodan, or as sister to trilobites. Multipodomerous uniramous deutocerebral antennae are known in nearly all other members of the clade (*Zhang et al., 2007*; *Du et al., 2019*; *Zhai et al., 2019*), and thus this feature is not very informative. Similarly, a conterminant hypostome was already known in numerous other artiopodans (*Zhang et al., 2004*; *Stein and Selden, 2012*). As this type of hypostomal attachment is considered ancestral within Trilobita (*Fortey, 1990*), its presence in the possible sister to trilobites (if dorsal sutures are considered homologous) is consistent with these expectations from trilobite anatomy. If *Acanthomeridion* is instead an early diverging artiopodan, the presence of natant hypostomes in both trilobitomorphs and vicissicaudates (*Fortey, 1990*; *Chen et al., 2019*) demonstrates that a natant hypostome has most likely evolved repeatedly within Artiopoda.

In summary, new fossils and CT-scan data reveal the ventral anatomy and appendages of *Acanthomeridion serratum* for the first time, and additional specimens demonstrate that *A. 'anacanthus'* represents a junior synonym of *A. serratum,* with some features thought to distinguish the two instead ontogenetic in origin. The presence of a posteriorly orientated spine on the cephalic ventral plates of *Acanthomeridion* might be homologous to the genal spine of trilobite librigenae, providing additional support for a.sister group relationship between these taxa. Phylogenetic analyses interrogating this hypothesis recover a sister relationship between *Acanthomeridion* and trilobites, but still require multiple origins of cephalic sutures within the group. If the ventral plates are instead treated as homologous to the doublure of trilobites, or as not homologous to any trilobite feature, then no close relationship is recovered, and the morphological similarities between *Acanthomeridion* and trilobites such as the path of the suture separating the ventral plates from the cephalon, and details of the ventral plates, should instead be treated as convergent. Thus, regardless of how the ventral plates of *Acanthomeridion* are interpreted, multiple origins of cephalic sutures in artiopodans are recovered by phylogenetic analyses.

## Materials and methods

New specimens of *Acanthomeridion* were collected from Jiucun town, Chengjiang (*Figure 2*) and accessioned at the Management Committee of the Chengjiang Fossil Site World Heritage (CJHMD 00052–00062), and Research Center of Paleobiology, Yuxi Normal University (YRCP 0016–0020). New specimens of *Wutingaspis tingi* (YRCP 0021) and *Xandarella spectaculum* (YRCP 0022) were collected from Haikou town, Kunming (*Figure 2*) and housed at Research Center of Paleobiology, Yuxi Normal University. The holotype of *Zhiwenia coronata* (YKLP 12370) is deposited at the Key Laboratory for Palaeobiology, Yunnan University which was excavated from Xiaoshiba area, Kunming (*Figure 2*). The specimens were photographed by a LEICA DFC 550 digital camera mounted to a Stereoscope LEICA M205 C, and scanned with a Zeiss Xradia 520 Versa X-ray Microscope and Nikon XTH 225 ST microfocus X-ray tomography machine. All micro-CT images were processed with the software Drishti (*Zhai et al., 2019*). Figures were processed with CorelDRAW X8, Inkscape 1.0, and Illustrator.

Given that specimens with thorax longitudinally curved and specimens with flattened thorax are considered as two species of *Acanthomeridion*, despite having the same exoskeletal morphology, we measured their axial length and attempted to determine their true maximum width in all specimens, including those described in this study and previously documented measurable specimens. The data were initially analyzed using a bivariate regression fitting function in Microsoft Office Excel. Subsequently, the software Past (*Hammer et al., 2001*) was employed for testing and validation, obtaining the same function and conducting a rational analysis of the function.

The character matrix used for the phylogenetic analyses was adapted from *Schmidt et al., 2022*, itself adapted from *Du et al., 2019* and *Chen et al., 2019*. *Olenellus getzi* was added, to facilitate comparison with olenelline trilobites, while *Anacheirurus adserai*, *Cryptolithus tesselatus,* and *Olenoides serratus* were added to provide more trilobite terminals. *Olenellus getzi* was chosen as it represents the olenelline with best known soft tissues (antennae only; *Dunbar, 1925*). Trunk appendages are known in *A. adserai, C. tesselatus,* and *O. serratus* (*Walcott, 1911*; *Whittington, 1975*; *Perez-Peris et al., 2021*), with further descriptions by *Bicknell et al., 2021*; *Losso and Ortega-Hernández, 2022* for coding the matrix. Our updated dataset includes 70 taxa (*Acanthomeridion anacanthus* was removed, four trilobites were added). Three matrices were used to test the alternative hypotheses of sutures, one with 100 characters considering the ventral plates unique, and not homologous to trilobite librigenae ('ventral plates under head' as a new character in the dataset) and two

with 99 characters. The first of these considered the ventral plates homologous to trilobite librigenae, and the second to the doublure of olenellines (see Morphobank project). We added the character state 'ringed distribution lamellae' to Ch. 15 in all matrices (*Chen et al., 2019*; *Du et al., 2019*), and altered the characters relating to the suture pattern (Ch. 31, 32) to include character states suitable for *Olenellus getzi*. Given the difficulties of identifying circumocular sutures in non-biomineralizing artiopodans, Ch. 31 indicated the presence or absence of ecdysial sutures in the cephalon, while Ch. 32 indicated the type of suture. This could either be marginal and/or circumocular, but unfused (state 0) or marginal and circumocular fused into a facial suture (state 1). This facilitated differentiation between the state in *Olenellus getzi* and the other trilobites in the dataset, and thus allowed the two hypotheses relating to homology between the ventral plates and the doublure or free cheeks of trilobites to be tested. Eight additional characters relating to the trilobite glabellar region were added, taken from *Paterson et al., 2019*, in order to provide data to distinguish internal trilobite relationships. These were Ch. 5, 6, 7, 10, 11, 12, 14, and 15 in *Paterson et al., 2019*, which are Ch. 37–44 in all three matrices used for this study. A further character (calcified thorax separated into prothorax and opisthothorax) was also added (Ch. 99 or 100 in matrices used for this study). Matrices used in phylogenetic analyses are provided through Morphobank Project 4290 (http://dx.doi.org/10.7934/P4290).

Phylogenetic analyses were performed with MrBayes (Bayesian Inference) (*Ronquist et al., 2012*). Both the 'maximum information' and 'minimum assumptions' strategies of ref (*Bapst et al., 2018*) were utilized on an earlier version of the matrix, in order to confirm that model choice was not a major driver in differences for the results (*Du et al., 2023*). For the final version, the 'maximum information' strategy was used, as model choice did not play a large influence on the topologies recovered. Bayesian analyses were run for 20 million generations using four chains, every 1000[th] sample stored and 25% burn-in (*Lewis, 2001*; *Ronquist et al., 2012*). Convergence was diagnosed using Tracer (*Rambaut et al., 2018*).

For each coding strategy, two sets of Bayesian analyses were run. The first was unconstrained, whereas for the second all trilobites except for *Olenellus getzi* were constrained as a monophyletic group to recover the relationships of trilobites (*Olenellus* as the earliest diverging member) comparable to the comprehensive trilobite phylogenetic analysis of *Paterson et al., 2019*. Resulting posterior tree samples were compared by comparisons in the consensus trees, and in multidimensional treespace (using R code and method from *Pates et al., 2022*). The number of trees where *Acanthomeridion serratum* was sister to trilobites was quantified for each coding strategy and for constrained and unconstrained analyses, and the area of treespace occupied by the posterior samples – and trees where *A. serratum* was sister to triobites within those samples – were visualized.

## Acknowledgements

The comments and suggestions of the editor and three anonymous reviewers greatly improved the manuscript. We thank Jian Han and Jie Sun for scanning the fossils and Javier Ortega-Hernández for helpful discussion.

## Additional information

### Funding

| Funder | Grant reference number | Author |
| --- | --- | --- |
| National Natural Science Foundation of China | 42262004 | Ai-lin Chen |
| National Natural Science Foundation of China | 42202003 | Kun-sheng Du |
| National Natural Science Foundation of China | 41662003 | Ai-lin Chen |
| State Key Laboratory for Palaeobiology and Stratigraphy | 193104 | Ai-lin Chen |

| Funder | Grant reference number | Author |
|--------|------------------------|--------|
| Herchel Smith Postdoctoral Fellowship | | Stephen Pates |
| Natural Environment Research Council | NE/X017745/1 | Stephen Pates |

The funders had no role in study design, data collection and interpretation, or the decision to submit the work for publication.

## Author contributions

Kun-sheng Du, Conceptualization, Data curation, Investigation, Visualization, Methodology, Writing – original draft, Writing – review and editing; Jin Guo, Resources, Data curation, Validation; Sarah R Losso, Conceptualization, Data curation, Formal analysis, Investigation, Visualization, Writing – original draft, Writing – review and editing; Stephen Pates, Conceptualization, Data curation, Formal analysis, Supervision, Validation, Visualization, Methodology, Writing – original draft, Project administration, Writing – review and editing; Ming Li, Data curation, Methodology; Ai-lin Chen, Resources, Supervision, Funding acquisition, Visualization, Project administration

## Author ORCIDs

Kun-sheng Du  https://orcid.org/0000-0002-3009-8323
Stephen Pates  https://orcid.org/0000-0001-8063-9469

Reviewer #1 (Public Review): https://doi.org/10.7554/eLife.93113.4.sa1
Reviewer #3 (Public Review): https://doi.org/10.7554/eLife.93113.4.sa2
Author response https://doi.org/10.7554/eLife.93113.4.sa3

# Additional files

## Supplementary files

• MDAR checklist

## Data availability

Datasets for the phylogenetic analyses are uploaded to Morphobank, Project 4290.

The following dataset was generated:

| Author(s) | Year | Dataset title | Dataset URL | Database and Identifier |
|-----------|------|---------------|-------------|-------------------------|
| Du KS, Guo J, Losso SR, Pates S, Li M, Chen AL | 2024 | Project 4290: Multiple origins of dorsal ecdysial sutures in trilobites and their relatives | http://dx.doi.org/10.7934/P4290 | Morphobank, 10.7934/P4290 |

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
