## [Editor Report · eLife assessment]

The authors present 16 new well-preserved specimens from the early Cambrian Chengjiang biota. These specimens potentially represent a new taxon which could be **useful** in sorting out the problematic topology of artiopodan arthropods - a topic of interest to specialists in Cambrian arthropods. The authors provide **solid** anatomical and phylogenetic evidence in support of a new interpretation of the homology of dorsal sutures in trilobites and their relatives.

---

## [Referee Report · Reviewer #1 (Public Review)]

Summary:

Du et al. report 16 new well-preserved specimens of atiopodan arthropods from the Chengjiang biota, which demonstrate both dosal and vental anatomies of a potential new taxon of atiopodans that are closely related to trolobites. Authors assigned their specimens to Acanthomeridion serratum, and proposed A. anacanthus as a junior subjective synonym of Acanthomeridion serratum. Critically, the presence of ventral plates (interpreted as cephalic liberigenae), together with phylogenic results, lead authors to conclude that the cephalic sutures originated multiple times within the Artiopoda.

Strengths:

New specimens are highly qualified and informative. The morphology of dorsal exoskeleton, except for the supposed free cheek, were well illustrated and described in detail, which provides a wealth of information for taxonomic and phylogenic analyses.

---

## [Referee Report · Reviewer #3 (Public Review)]

Summary:

Well-illustrated new material is documented for Acanthomeridion, a formerly incompletely known Cambrian arthropod. The formerly known facial sutures are proposed be associated with ventral plates that the authors homologise with the free cheeks of trilobites (although also testing alternative homologies). An update of a published phylogenetic dataset permits reconsideration of whether dorsal ecdysial sutures have a single or multiple origins in trilobites and their relatives.

Strengths:

Documentation of an ontogenetic series makes a sound case that the proposed diagnostic characters of a second species of Acanthomeridion are variation within a single species. New microtomographic data shed light on appendage morphology that was not formerly known. The new data on ventral plates and their association with the ecdysial sutures are valuable in underpinning homologies with trilobites.

---

## [Author Response]

The following is the authors’ response to the previous reviews.

**Public Reviews:**

**Reviewer #1 (Public Review):**
Summary:Du et al. report 16 new well-preserved specimens of atiopodan arthropods from the Chengjiang biota, which demonstrate both dosal and vental anatomies of a pothential new taxon of atiopodans that are closely related to trolobites. Authors assigned their specimens to Acanthomeridion serratum, and proposed A. anacanthus as a junior subjective synonym of Acanthomeridion serratum. Critially, the presence of ventral plates (interpreted as cephalic liberigenae), together with phylogenic results, lead authors to conclude that the cephalic sutures originated multiple times within the Artiopoda.Strengths:New specimens are highly qualified and informative. The morphology of dorsal exoskeleton, except for the supposed free cheek, were well illustrated and described in detail, which provide a wealth of information for taxonmic and phylogenic analyses.Weaknesses:The weaknesses of this work is obvious in a number of aspects. Technically, ventral morphlogy is less well revealed and is poorly illustrated. Additional diagrams are necessary to show the trunk appendages and suture lines. Taxonomically, I am not convinced by authors' placement. The specimens are markedly different from either Acanthomeridion serratum Hou et al. 1989 or A. anacanthus Hou et al. 2017. The ontogenetic description is extremely weak and the morpholical continuity is not established. Geometric and morphomitric analyses might be helpful to resolve the taxonomic and ontogenic uncertainties. I am confused by author's description of free cheek (libragena) and ventral plate. Are they the same object? How do they connect with other parts of cephalic shield, e.g. hypostome and fixgena. Critically, homology of cephalic slits (eye slits, eye notch, doral suture, facial suture) not extensivlely discussed either morphologically or functionally. Finally, authors claimed that phylogenic results support two separate origins rather than a deep origin. However, the results in Figure 4 can be explain a deep homology of cephalic suture in molecular level and multiple co-options within the Atiopoda.Comments on the revised version:I have seen the extensive revision of the manuscript. The main point "Multiple origins of dorsal ecdysial sutures in atiopoans" is now partially supported by results presented by the authors. I am still unsatisfied with descriptions and interpretations of critical features newly revealed by authors. The following points might be useful for the author to make further revisions.(1) The antennae were well illustrated in a couple of specimens, while it was described in a short sentence.

Some more details of the changing article shape and overall length of antennae has been added to the description.

(2) There are also imprecise descriptions of features.

Measurements, dimensions and multiple figures are provided for many features in the text and the supplement includes more figures. In total, 11 figures are provided with details (photographs or measurements) of the material.

(3) Ontogeny of the cephalon was not described.

A sentence has been added to the description to note the changing width:length of the cephalon during ontogeny, with a reference to Figure 6.

(3) The critical head element is the so called "ventral plate". How this element connects with the cephalic shield is not adequately revealed. The authors claimed that the suture is along the cephalic margin. However, the lateral margin of cephalon is not rounded but exhibit two notches (e.g. Fig 3C) . This gives an indication that the supposed ventral plates have a dorsal extension to fit the notches. Alternatively, the "ventral plate" can be interpreted as a small free cheek with a large ventral extension, providing evidence for librigenal hypothesis.

As noted in the diagnosis for the genus, these notches are interpreted to accommodate the eye stalks. The homology of the ventral plates is discussed at length in the manuscript, and is the focus of the three sets of phylogenetic analyses performed.

**Reviewer #3 (Public Review):**
Summary:Well-illustrated new material is documented for Acanthomeridion, a formerly incompletely known Cambrian arthropod. The formerly known facial sutures are proposed be associated with ventral plates that the authors homologise with the free cheeks of trilobites (although also testing alternative homologies). An update of a published phylogenetic dataset permits reconsideration of whether dorsal ecdysial sutures have a single or multiple origins in trilobites and their relatives.Strengths:Documentation of an ontogenetic series makes a sound case that the proposed diagnostic characters of a second species of Acanthomeridion are variation within a single species. New microtomographic data shed light on appendage morphology that was not formerly known. The new data on ventral plates and their association with the ecdysial sutures are valuable in underpinning homologies with trilobites.I think the revision does a satisfactory job of reconciling the data and analyses with the conclusions drawn from them. Referee 1's valid concerns about whether a synonymy of Acanthomeridion anacanthus is justified have been addressed by the addition of a length/width scatterplot in Figure 6. Referee 2's doubts about homology between the librigenae of trilobites and ventral plates of Acanthomeridion have been taken on board by re-running the phylogenetic analyses with a coding for possible homology between the ventral plates and the doublure of olenelloid trilobites. The authors sensibly added more trilobite terminals to the matrix (including Olenellus) and did analyses with and without constraints for olenelloids being a grade at the base of Trilobita. My concerns about counting how many times dorsal sutures evolved on a consensus tree have been addressed (the authors now play it safe and say "multiple" rather than attempting to count them on a bushy topology). The treespace visualisation (Figure 9) is a really good addition to the revised paper.Weaknesses:The question of how many times dorsal ecdysial sutures evolved in Artiopoda was addressed by Hou et al (2017), who first documented the facial sutures of Acanthomeridion and optimised them onto a phylogeny to infer multiple origins, as well as in a paper led by the lead author in Cladistics in 2019. Du et al. (2019) presented a phylogeny based on an earlier version of the current dataset wherein they discussed how many times sutures evolved or were lost based on their presence in Zhiwenia/Protosutura, Acanthomeridion and Trilobita. The answer here is slightly different (because some topologies unite Acanthomeridion and trilobites). This paper is not a game-changer because these questions have been asked several times over the past seven years, but there are solid, worthy advances made here.I'd like to see some of the most significant figures from the Supplementary Information included in the main paper so they will be maximally accessed. The "stick-like" exopods are not best illustrated in the main paper; their best imagery is in Figure S1. Why not move that figure (or at least its non-redundant panels) as well as the reconstruction (Figure S7) to the main paper? The latter summarises the authors' interpretation that a large axe-shaped hypostome appears to be contiguous with ventral plates.

We have moved these figures from the supplementary information to the main text, and renumbered figures accordingly. Fig S1 has now been split – panels a and b are in the main text (new Fig. 4), with the remainder staying as Fig S1. Fig S7 is now Fig. 8 in the main text.

The specimens depict evidence for three pairs of post-antennal cephalic appendages but it's a bit hard to picture how they functioned if there's no room between the hypostome and ventral plates. Also, a comment is required on the reconstruction involving all cephalic appendages originating against/under the hypostome rather the first pair being paroral near the posterior end of the hypostome and the rest being post-hypostomal as in trilobites.

A short comment has been added to the caption.

**Recommendations for the authors:**

**Reviewer #1 (Recommendations For The Authors):**
I have seen the extensive revision of the manuscript. The main point "Multiple origins of dorsal ecdysial sutures in atiopoans" is now partially supported by results presented by the authors. I am still unsatisfied with descriptions and interpretations of critical features newly revealed by authors. The following points might be useful for the author to make further revisions.(1) The antennae were well illustrated in a couple of specimens, while it was described in a short sentence.(2) There are also imprecise descriptions of features (see my annotations in submitted ms).(3) Ontogeny of the cephalon was not described.(3) The critical head element is the so called "vental plate". How this element connects with the cephalic shield is not adequately revealed. The authors claimed that the suture is along the cephalic margin. However, the lateral margin of cephalon is not rounded but exhibit two notches (e.g. Fig 3C) . This gives a indication that the supposed ventral plates have a dorsal extension to fit the notches. Alternatively, the "ventral plate" can be interpreted as a small free cheek with a large ventral extension, providing evidence for librigenal hypothesis.
**Reviewer #3 (Recommendations For The Authors):**
The references swap back and forth between journal titles being abbreviated or written out in full. Please standardise this to journal format rather than alternating between two different styles.Line 145: Perez-Peris et al. (2021) should be cited as the source for the Anacheirurus appendages.

Added, thank you.

Line 310: The El Albani et al (2024) paper on ellipsocephaloid appendages should be noted in connection with an A+4 (rather than A+3) head in trilobites.

Added.

Minor or trivial corrections:Line 51: move the three citations to follow "arthropods" rather than following "artiopodans", as none of these papers are specifically about Artiopoda.

Changed thank you

Caption to Figure 1 and line 100: Acanthomeridion appears in Figure 1 and in the text with no context. Please weave it into the text appropriately.Line 136: The data were...

Corrected

Line 164: upper case for Morphobank.

Corrected

Line 183: spelling of "Village" (not "Vallige").

Corrected

Line 197: I suggest using "articles" rather than "podomeres" for the antenna (as you did in line 232).

Changed thank you

Line 269: "gnathobasal spine (rather than "spin").

Changed thank you

Line 272: "Exopods" is used here but elsewhere "exopodites" is used.

Exopodites is now used throughout

Line 359: "can been seen" is awkward and, as evolutionary patterns are inferred rather than "seen", could be reworded as "... loss of the eye slit has been inferred...".

Reworded as suggested

Line 422 and 423: As two referees asked in the first round of review, delete "iconic" and "symbolic".

Deleted as suggested

Line 467: "librigena-like".

Corrected